# Quantifying the Detrimental Effects of Multiple Freeze/Thaw Cycles on Primary Human Lymphocyte Survival and Function

**DOI:** 10.3390/ijms24010634

**Published:** 2022-12-30

**Authors:** Valentina Serra, Edoardo Fiorillo, Francesco Cucca, Valeria Orrù

**Affiliations:** 1Institute for Genetic and Biomedical Research, National Research Council (CNR), 08045 Lanusei, Italy; 2Department of Biomedical Sciences, University of Sassari, 07100 Sassari, Italy

**Keywords:** PBMC cryopreservation, viability, immunophenotypes, freezing cycles, flow cytometry, cell function

## Abstract

The use of cryopreserved peripheral blood mononuclear cells is common in biological research. It is widely accepted that primary cells are rendered unusable by several freezing cycles, although this practice might be very helpful when the biological material is valuable and its re-collection is impractical. To determine the extent to which primary cells undergoing repeated freezing cycles are comparable to one another and to fresh samples, we evaluated overall lymphocyte viability, their proliferation and cytokine production capabilities, as well as the levels of 27 cell subtypes in ten human peripheral blood mononuclear cells frozen for five years and repeatedly thawed. As expected, we observed a progressive increase in cell death percentages on three rounds of thawing, but the frequency of the main lymphocyte subsets was stable across the three thawings. Nevertheless, we observed a significant reduction of B cell frequency in frozen samples compared to fresh ones. On repeated thawings and subsequent conventional stimulation, lymphocyte proliferation significantly decreased, and IL-10, IL-6, GM-CSF, IFN-gamma, and IL-8 showed a trend to lower values.

## 1. Introduction

In recent years, the implication of specific immune cells in disease predisposition was identified by searching for genetic signals simultaneously associated with complex diseases and immune cells through genome wide association studies [1,2,3,4]. Importantly, immune profiling for these genetic analyses were mostly dissected in population cohorts by flow cytometry. In some cases, the same genetic variant affected both the level of an immune cell type and the risk of a disease, supporting a causal relationship [1]. For instance, a variant located in the *TNFSF13B* gene, regulating the levels of B cells in the general population, was also a disease-predisposing variant for multiple sclerosis and systemic lupus erythematosus (SLE), its mechanism of action was characterized, thereby the role of B cells in the diseases was clarified [5]. Another study, using an approach based on phenotypic comparisons between patients and controls, has identified activated naïve B cells and DN2 subset of IgD−CD27− B cells as precursors of autoantibody producing plasma cells in SLE [6]. Thus, the identification of a link between specific immune cells and complex diseases is an active and expanding area of research.

To ensure high data quality and the reproducibility of immune profiling, stringent rules must be followed, ranging from the choice of sample type to the sample size, and the standardization of protocols. In this context, cryopreservation of human peripheral blood mononuclear cells (PBMCs) is a common procedure across laboratories to preserve high cell viability, reducing inter-assay variability, and ensure good quality of immunophenotyping, although some antigens and cell types have been shown to be damaged [7]. Recently, ours and other groups showed that frozen whole blood can also be used for immunophenotyping [8,9,10].

In laboratory practice, it has been accepted that PBMCs cannot undergo several freeze/thaw cycles, based on the common assumption that repeated cryopreservation would likely induce significant changes in lymphocyte phenotype and function [11]. However, there have been no studies demonstrating that this practice would damage primary cells or assessing the extent to which freezing cycles may alter cell viability and function. As the use of repeatedly frozen and thawed cells would be very useful for samples that are valuable and hard to recollect, as in a human bioresource, we have directly assessed the viability, functionality and immunophenotypic characteristics of PBMCs undergoing several freezing cycles compared to the initially frozen cells.

We measured overall viability, proliferation activity, cytokine production, and levels of 27 lymphocyte subsets in ten samples collected in 2017 and stored for five years at −150 °C. We then extended the study to the same samples exposed to two further freezing cycles and used the immune profiling performed on the original fresh samples as a standard for comparisons. Our results showed that although viability and proliferation activity were significantly affected by multiple freezing cycles, the frequency of the major lymphocyte subsets was comparable after repeated thawings.

## 2. Results

### 2.1. Multiple Freezing Cycles Affect Lymphocyte Viability but Not Cell Frequencies

As expected, freezing cycles significantly affected lymphocyte viability. Indeed, lymphocyte viability significantly decreased after repeated freezing cycles, with average values of 94%, 87% and 79% after the first, second, and third thawing, respectively (*p*-value 1st vs. 2nd thawing = 4.83 × 10^−5^ and *p*-value 1st vs. 3rd thawing = 2.70 × 10^−9^) (Figure 1 and Figure 2 and Appendix A).

We then measured the level of 27 immune cell subsets, including B and T cells and their maturation stages, in ten PBMC samples subjected to multiple freezing cycles. As a reference control for immune profiling, we used the frequencies of the same 27 cell types measured in fresh blood from the same 10 individuals, profiled in 2017.

Interestingly, the frequencies of the assessed subsets were not affected by multiple thawing procedures; furthermore, they were comparable to those observed in freshly processed blood, apart from total B cells, which showed a significant reduction in frozen samples compared to fresh blood (*p*-value = 5 × 10^−7^) (Figure 3 and Appendix A), in line with previous observations [9,12,13]. Importantly, B cell subset frequencies remained stable.

In contrast to previous reports [13], we did not observe a significant increase of T cell frequencies in frozen with respect to fresh samples.

### 2.2. Multiple Freezing Cycles Affect Proliferation Activity but Not Cytokine Production

To assess proliferation activity and cytokine production, we stimulated PBMCs with anti-CD3 and anti-CD28 antibodies (Figure 4). Multiple freezing cycles significantly reduced lymphocyte proliferation capabilities, with average proliferation rates of 63%, 48% and 39% for samples thawed once, twice, and three times, respectively (*p*-value = 7.90 × 10^−3^ and *p*-value = 1.69 × 10^−5^, respectively) (Figure 5, Appendix A). We also assessed 10 cytokines: IL-1 beta, IL-10, IL-6, GM-CSF, IL-5, IFN-gamma, TNF-alpha, IL-2, IL-4, and IL-8 at basal level and after stimulation. Eight of ten cytokines significantly increased after stimulation, indicating that cells were functional and responsive (Appendix A). Nevertheless, we observed a downward trend for IL-10, IL-6, GM-CSF, IFN-gamma, and IL-8 with increasing freezing cycles (Figure 6) consistent with a reduction of cell functionality during multiple freezing cycles.

## 3. Discussion

PBMC cryopreservation is a standardized and reliable method, widely used in research field, that allows preservation of cell viability and functionality for subsequent analysis. PBMCs fulfill a variety of experimental needs, particularly when the sampling period spans for a long time or when collection centers are located in different areas. In this context, PBMC collection and storage allow simultaneous analysis of large batches of samples, substantially reducing inter-assay variability. Of course, biobanking of frozen cells has a much greater range of use in clinical practice, including storage for reproductive use or tissue and organ regeneration. Thus, we anticipated that better characterization of cryopreserved cells after multiple freezing cycles could have general utility beyond basic research.

During currently optimized cell freezing procedures, cell movement and biochemical processes are stopped by slowly reducing cell water temperature avoiding cell damage [14]. Furthermore, the use of a cryoprotectant such as dimethyl sulfoxide (DMSO) prevents formation of intracellular and extracellular crystals during the freezing process, thus further helping to preserve cellular integrity. Thawing of cells, while much faster, also comprises several steps, optimized across laboratories, to maintain high cell recovery and viability [15].

Nonetheless, PBMC cryopreservation and thawing are relative violent processes that can potentially induce changes in cell phenotype and functions [11]. Consequently, primary cells are generally only subjected to one freezing cycle, and multiple freezing steps are not traditionally considered applicable in research practice. However, we have found no reports that describe the effects of repeated freezing and thawing cycles on cells. To provide information about these effects, we assessed viability, phenotype, and function of PBMCs after up to three consecutive cryopreservation cycles. We observed that multiple thawings affected lymphocyte viability, gradually increasing the proportion of dead cells as the number of thaws increased. Notably, most of the cells remained viable, and unexpectedly, their major phenotypes are comparable after repeated freezing cycles (Appendix A). Preliminary studies, such as the present work, can identify cell populations that could be reliably dissected even after multiple rounds of cryopreservation, and correspondingly, highlight cell types–notably B cells-for which the process is detrimental to many possible follow-up studies. The availability of the same cell samples after more than one freeze/thaw cycle raises the possibility that the same samples could be used for further phenotypic assessments; but caution is counseled by our observations of a reduction in proliferation activity and a downward trend for cytokine production capability after repeated freeze/thawing cycles.

In a likely extension of the approach used here, it could be useful to assess if and to what extent multiple freezing cycles affect myeloid compartment cells, again evaluating cell types and their functionality.

Additionally, further experiments can explore whether a longer time between thawing cycles may add to variability and damage, and more specific assays, to evaluate suppression activity or phosphorylation status, could be tested to assess functionality of lymphocytes subsets such as regulatory and helper T cells.

## 4. Materials and Methods

### 4.1. The SardiNIA Dataset

SardiNIA project cohort is a longitudinal study comprising about 8000 general population volunteers (57% females, 43% males), ranging from 18 to 102 years and living in the central east coast of Sardinia, Italy. Ten samples, (6 females and 4 males) ranging from 34 to 66 years, from the SardiNIA project bioresource were randomly selected and used in this study to assess viability and immunophenotyping; nine out of ten samples were also used to evaluate proliferation capacity and cytokine production after three freezing cycles.

### 4.2. Peripheral Blood Mononuclear Cell Isolation and Cryopreservation

PBMCs were isolated in 2017 from whole blood collected in BD Vacutainer heparinized blood collection tubes (BD Life Sciences, Franklin Lakes, NJ, USA, #367878) by histopaque density gradient medium.

In detail, whole blood was firstly diluted with Hank’s balanced salt solution, and slowly dispensed on Hystopaque 1077, in a blood:Hank’s:Hystopaque ratio of 1:1:1. The resulting samples were centrifuged at 550× *g* for 30 min at room temperature. PBMC layer was then aspirated and washed twice, the first time with Phosphate Buffered Saline 1X (PBS 1X) and the second time with complete RPMI 1640 medium (supplemented with 10% of Fetal bovin Serum-FBS, Sodium Pyruvate and Glutamine). Samples were centrifuged at 250× *g* for ten minutes at room temperature.

After isolation, PBMCs (diluted with complete RPMI at about 1 × 10^7^ cells per mL) were frozen by adding a freezing media consisting of 20% DMSO diluted in FBS. The ratio of blood to freezing media was 1:1, thus the final concentration of DMSO was 10%. Samples were stored at −80 °C by using the Mr Frosty container (ThermoFisher Scientific, Waltham, MA, USA) to achieve the optimal cell cooling rate of −1 °C/minute, then transferred at −150 °C until first thawing.

### 4.3. Thawing, Cell Population Assessment and PBMC Refreezing

PBMCs were quickly thawed at 37 °C. 200 µL of the suspension, consisting of cells and freezing media, were washed once with PBS 1X, stained with the eleven fluorochrome conjugate antibodies specified in Appendix A, incubated for 30 min at +4 °C and analyzed by the Cytek Aurora analyzer (Cytek Biosciences, Fremont, CA, USA).

Briefly, cell viability was measured by using 7-aminoactinomycin D (7-AAD), an intercalating dye that can pass through the compromised cell membrane and bind the DNA, thus live cells will be 7-AAD negative.

B cells were characterized for their maturation stages using IgDvsCD27 and IgDvsCD38 markers; the expression of immunoglobulin A and D were also assessed [16].

Helper (CD4+) and cytotoxic-enriched T cells (CD8e) were characterized for their maturation stages by the expression of the phosphatase isoform CD45RA and the chemokine receptor CCR7 [17,18]. See Figure 1 for details.

Overall, lymphocyte viability and 27 cell frequencies with respect to hierarchically higher cells population were assessed.

The remaining cell suspension was frozen by using the Mr Frosty container to achieve the temperature of −80 °C, then transferred at −150 °C. Thawing and freezing procedures was repeated twice for each sample.

Fresh sample characterization and cell frequencies are from Orrù et al., 2020 [1].

### 4.4. Cytek Aurora Settings

The eleven-colour panel was assessed by the Cytek Aurora full spectrum cytometer (Cytek Biosciences) equipped with five lasers and 64 detectors. Unlike conventional cytometry that measures the peak of emission of each fluorochrome, spectral cytometry can measure the full emission spectra of a fluorophore across all laser lines.

To ensure instrument setting stability, the QC setup was performed daily by the SpectroFlo QC beads (Cytek Biosciences), enabling comparison of experiments conducted in different days. The SpectroFlo QC beads are 3-μm polystyrene microspheres highly fluorescent in all 64 detectors of the Aurora cytometer. During the QC, laser delays and area scaling factors were optimized and settings adjusted so that the fluorescence of beads reached the mean fluorescence intensity target values, accounting for day-to-day instrument variability.

### 4.5. Proliferation Assay and Cytokine Production

PBMCs were quickly thawed at 37 °C and washed with PBS 1X. 5 × 10^5^ cells resuspended in 500 µL of PBS 1X were labeled with 1 µM carboxyfluorescein diacetate succinimidyl ester (CFSE) and incubated for 15 min at 37 °C. Samples were then washed with complete RPMI medium. 2.5 × 10^5^ lymphocytes were stimulated in 150 µL of complete media with 2.5 × 10^5^ anti-CD3/anti-CD28 Dynabeads (Life Technologies, Carlsbad, CA, USA cat. 11131D), or left untreated, for 72 h at 37 °C with 5% CO_2_. After incubation, supernatants were harvested and stored at −80 °C for subsequent cytokine analysis. Cells were then washed with PBS 1X and stained with anti-CD45 and 7-AAD (Appendix A) for 30 min in the dark. Samples were finally washed with PBS 1X and analyzed by the FACS CANTO II cytometer (BD Biosciences, San Jose, CA, USA).

Cytokine production was assessed on cell supernatants by using Luminex multiplex technology and the Bio-Plex MAGPIX Multiplex Reader (BioRad, Hercules CA, USA). The human cytokine magnetic 10-plex panel (Life Technologies LHC0001M) were used to quantify human granulocyte-macrophage colony-stimulating factor (GM-CSF), interferon- γ (IFN-γ), interleukin (IL) 1β (IL-1β), IL-2, IL-4, IL-5, IL-6, IL-8, IL-10, and Tumor Necrosis Factor α (TNF-α), according to manufacturer’s instructions. To assess cell function, the difference in cytokine production after stimulation and at baseline was calculated, this difference was also compared among different freezing cycles.

### 4.6. Data Analysis

Cell characterization was carried out on fresh blood and on frozen PBMCs subjected to three thawing cycles after five years from the day of cryopreservation. Flow cytometry data was manually gated by the SpectroFlo software (Cytek Biosciences) according to the gating strategy described in Figure 1 and by the BD FACS DIVA software version 9.0 (BD Biosciences) for Figure 4.

Lymphocyte viability, cell frequencies, lymphocyte proliferation, and cytokine production were measured after each thawing cycle and compared to each other by using the paired *t*-test. Due to the small sample size, we also applied the Wilcoxon Mann Whitney non-parametric test. For both tests, a *p*-value < 0.05 was considered as statistically significant. For simplicity, we only indicated the *p*-value derived from the paired t-test along the main text. *p*-values from both tests were indicated in Appendix A.

Using the same statistical approach, we also compared the average cell frequencies measured on fresh sample (whole blood processed in 2017) with those measured after the three thawing cycles.

## Figures and Tables

**Figure 1 ijms-24-00634-f001:**
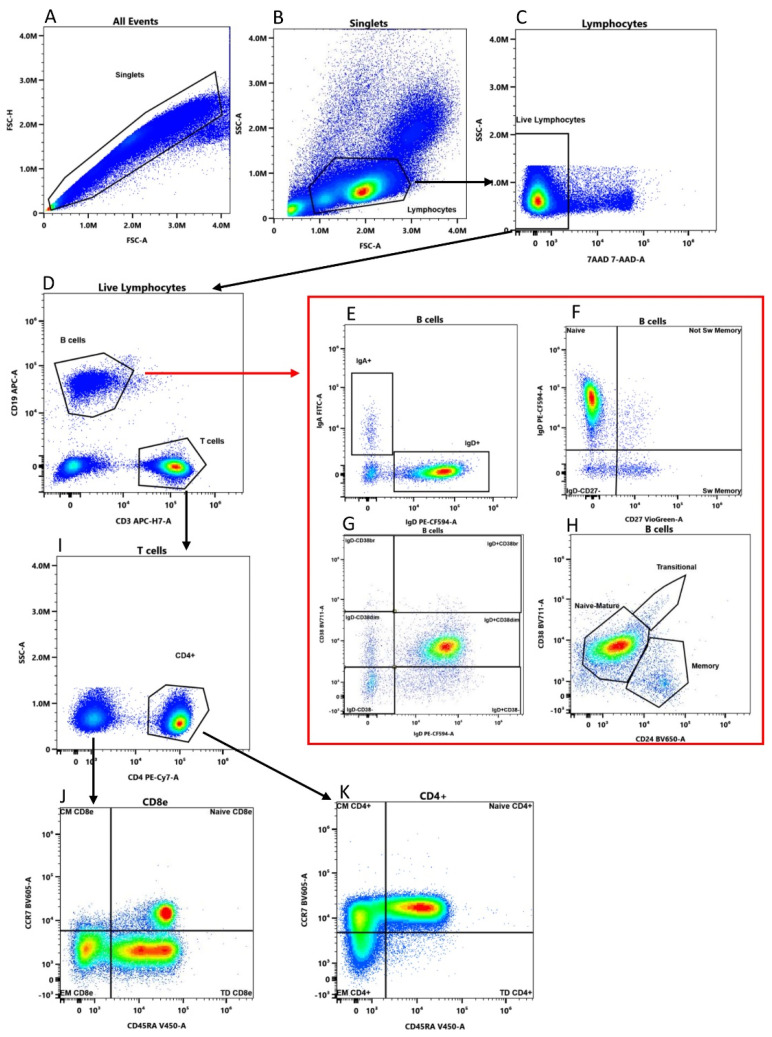
Gating strategy of a representative sample. (**A**,**B**) Lymphocytes were identified based on morphological parameters. (**C**) 7-AAD negative single cells were considered viable lymphocytes and (**D**) divided in B (CD19+ CD3− and T cells (CD19− CD3+). (**E**) B cells (red frame) were divided in IgD+ and IgA+ B cells; (**F**) switched memory (CD27+ IgD−), un-switched memory (CD27+ IgD+), naïve (CD27− IgD+) and CD27− IgD− B cells; (**G**) IgD+ CD38−, IgD+ CD38dim, IgD+ CD38br, IgD− CD38br, IgD− CD38dim, and IgD− CD38− subsets. (**H**) CD24 and CD38 define transitional (CD24+ CD38hi), memory (CD24+ CD38−/dim) and naïve-mature (CD24− CD38−/dim) subsets. (**I**) T cells not expressing CD4 were considered CD8-enriched cells (CD8e). (**J**) CD8e and (**K**) CD4+ T cells were divided in Central Memory (CM, CD45RA− CCR7+), Naïve (CD45RA+ CCR7+), Effector Memory (EM, CD45RA− CCR7−) and Terminally Differentiated (TD, CD45RA+ CCR7−).

**Figure 2 ijms-24-00634-f002:**
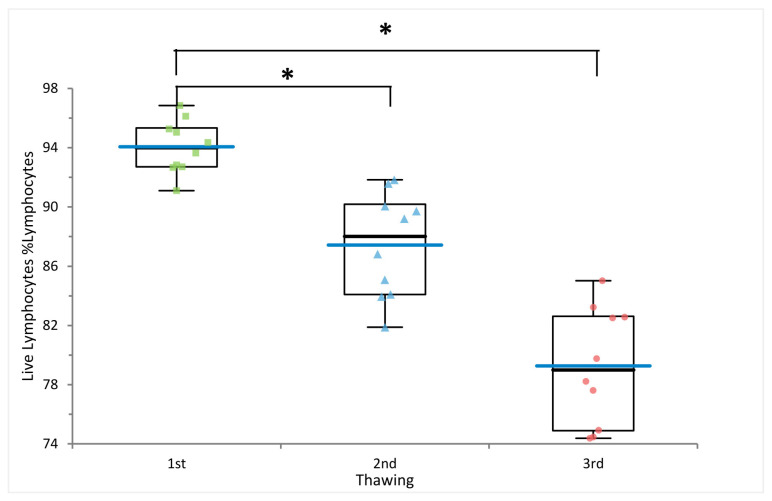
Lymphocyte viability. Comparison of lymphocyte viability measured on ten PBMC samples after the first, second, and third freezing cycle. * *p*-values from the paired *t*-test < 0.05. Blue bars indicate mean values.

**Figure 3 ijms-24-00634-f003:**
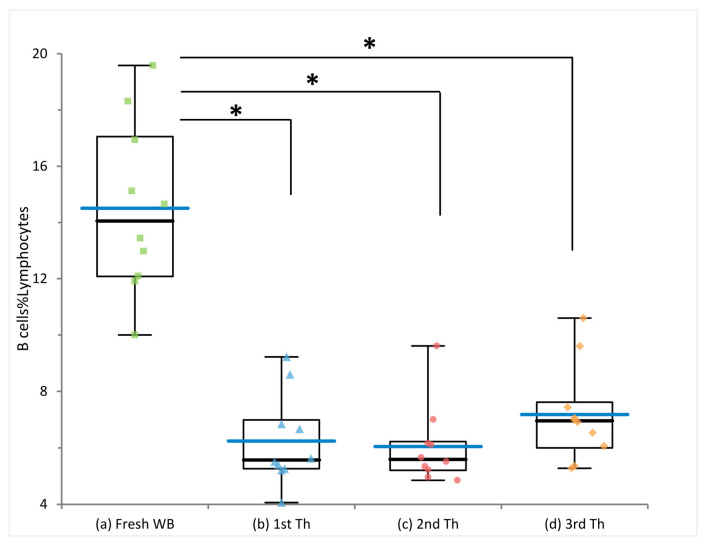
Effect of cryopreservation and multiple freezing cycles on B cell frequency. Comparison of B cell frequency measured in fresh whole blood (WB) and in PBMC subjected to multiple thawing (Th) cycles. * *p*-values from the paired *t*-test < 0.05. Blue bars indicate mean values.

**Figure 4 ijms-24-00634-f004:**
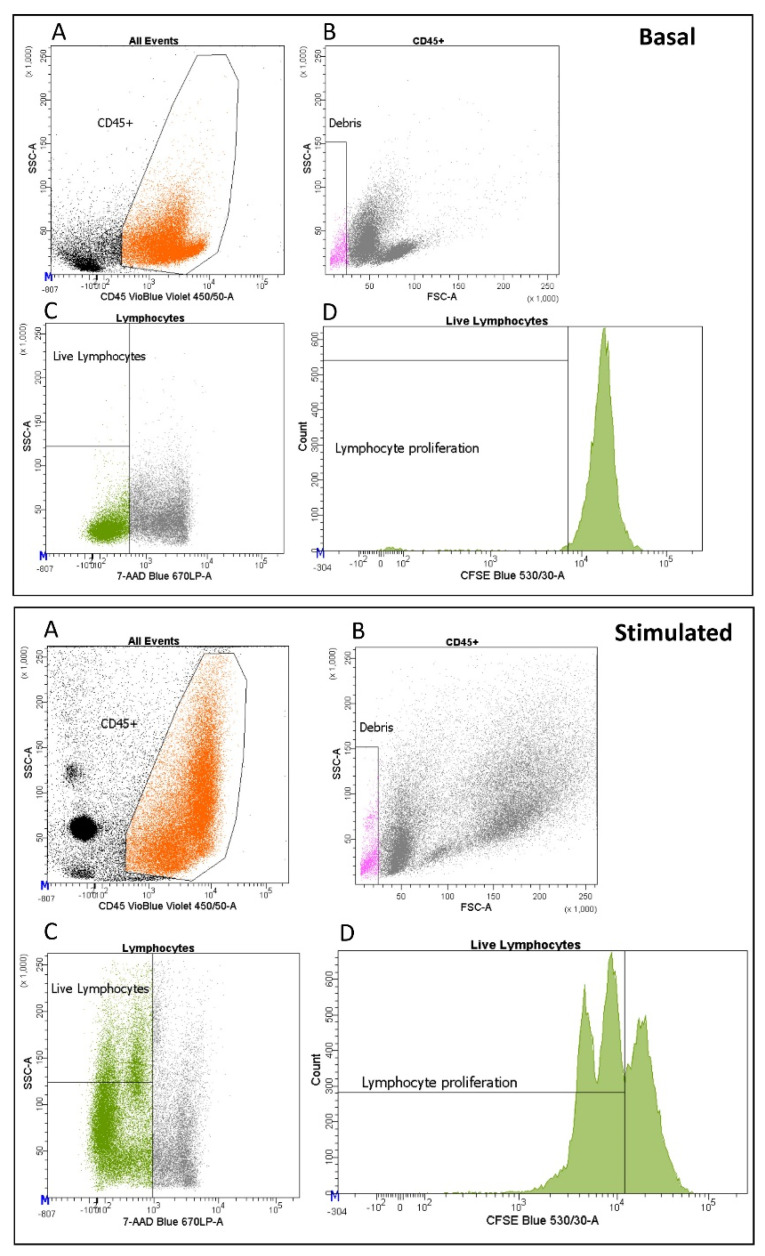
Proliferation activity of a representative sample. (**A**) Lymphocytes (orange) were identified by morphological parameter and CD45 positivity. (**B**) Debris (pink) were removed. (**C**) The remaining 7-AAD negative events correspond to live lymphocytes (green). (**D**) Proliferation was assessed by CFSE (carboxyfluorescein diacetate succinimidyl ester) at basal level and after stimulation.

**Figure 5 ijms-24-00634-f005:**
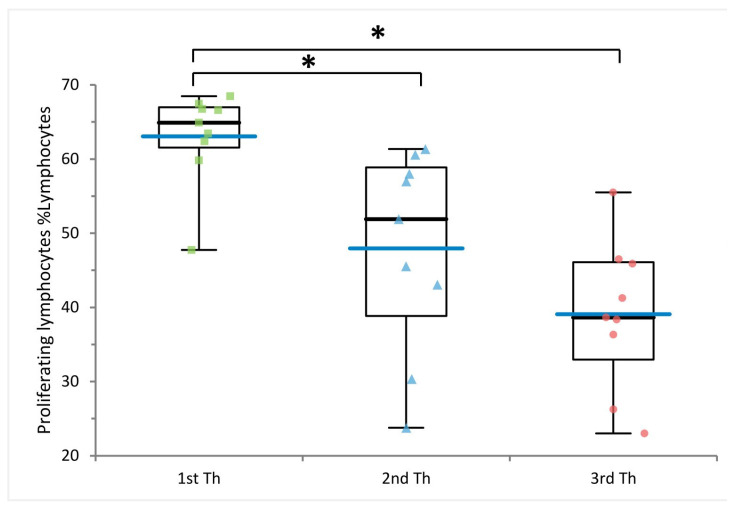
Effect of multiple freezing cycles on lymphocyte proliferation. Comparison of lymphocyte proliferation rate measured on nine PBMC samples after the first, second and third thawing (Th). * *p*-values from the paired *t*-test <0.05. Blue bars indicate mean values.

**Figure 6 ijms-24-00634-f006:**
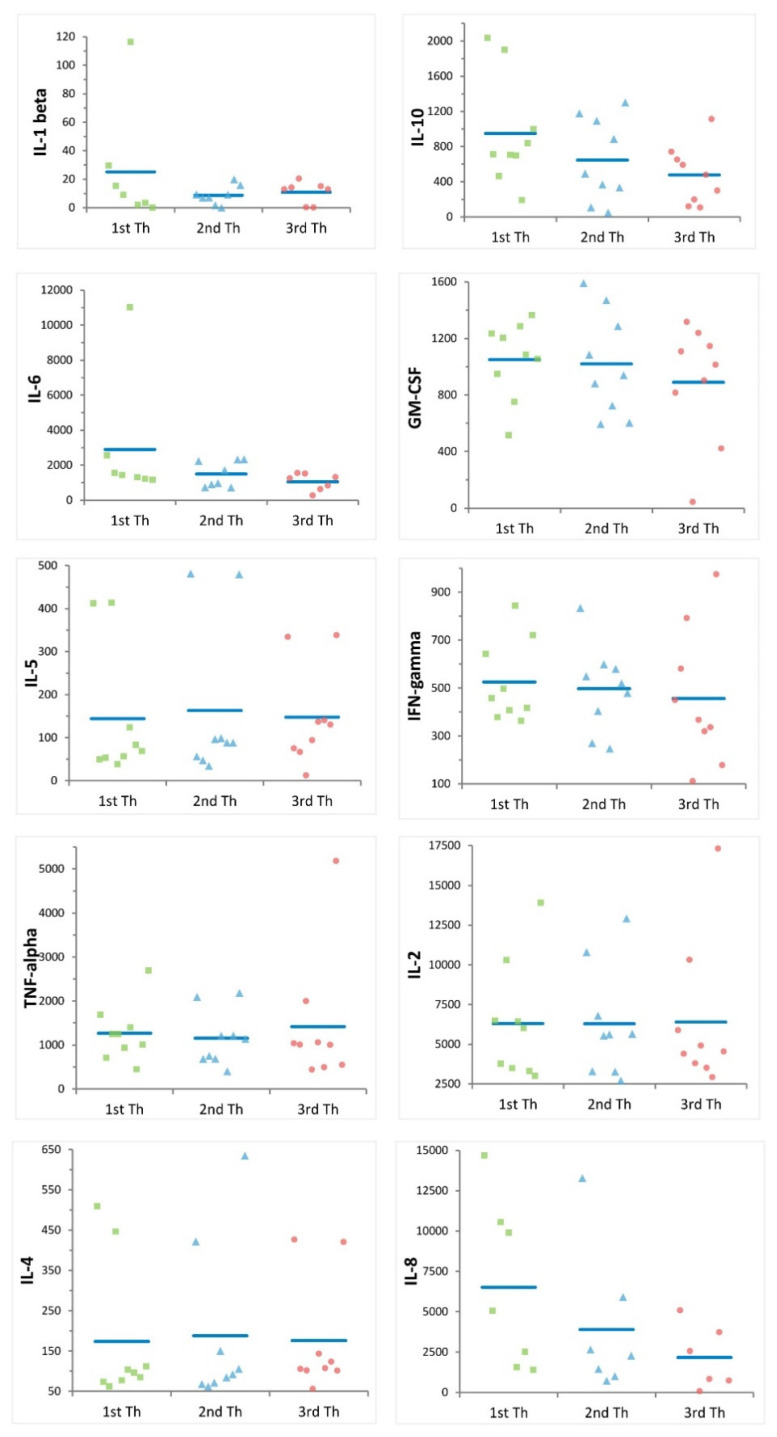
Effect of multiple freezing cycles on cytokine production. Each plot shows the cytokine production (expressed as difference between their concentration after stimulation and at basal level, see Section 4 in nine samples after one (green), two (blue) and three (red) thawing steps (Th). Blue bars indicate mean values.

## Data Availability

Not applicable.

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
