# Peer review of "Quantifying the Detrimental Effects of Multiple Freeze/Thaw Cycles on Primary Human Lymphocyte Survival and Function"

_ijms, 2022, doi:10.3390/ijms24010634_

Round 1
Reviewer 1 Report
The authors assessed the viability, functionality and immunophenotypic characteristics of peripheral blood mononuclear cells (PBMCs) undergoing 42 several freezing cycles compared to the initially frozen cells. The experimental design, methodology, data collection and interpretation are all sounds good. The suggestion from the reviewer is that: 1) The Supplementary Figure 1-3 are very important for readers to understand the whole article, so it's better to include them directly in the text as formal figures; 2) English language and style are fine, but minor check is needed.
Reviewer 2 Report
The study presents novel information that is very valuable to researchers conducting studies with PBMCs. The results are clearly presented and rationale is sound. There are couple of areas that the manuscript could be improved prior to publication.
In the methods section, please include details on who the samples were collected from (i.e., age/sex). How were these samples chosen (was it random?).
Much of the first paragraph of discuss (lines 95-115) are more appropriate for the introduction and do not really address the present findings. Including this information in the introduction and providing a more concise discussion of the findings would greatly improve the overall manuscript.
A couple of minor comments below:
Lines 36-37: please provide reference for this statement.
Line 50: unclear what is meant by "was essentially completed maintained". Please revise.
